# Analysis of Regional Division of Labor in Value Chain Patterns and Driving Factors in the Yangtze River Delta Region Using the Electronic Information Manufacturing Industry as an Example

**Jiangjiang Kang** [1,*], **Chuankai Yang** [2] **and Yuemin Ning** [3]

1 Institute of Applied Economics, Shanghai Academy of Social Sciences, Shanghai 200020, China
2 Institute of Urban and Demographic Studies, Shanghai Academy of Social Sciences, Shanghai 200020, China; yck@sass.org.cn
3 School of Geographic Sciences, East China Normal University, Shanghai 200241, China
* Correspondence: jjkang@sass.org.cn

**Abstract:** The electronic information manufacturing industry is characterized by a very significant intra-product specialization and can display the characteristics of a regional division of labor. Looking at the existing literature, most studies have mainly examined the position of different countries in the spatial division of labor from the perspective of global value chains, with fewer empirical analyses at the city level or regional scale. Furthermore, deepening the regional division of labor in value chains is an effective way to promote regional industrial synergy and high-quality economic development. Based on the number of listed enterprises and the total number of parent–subsidiary investment connections in the electronic information manufacturing industry, this study reveals the characteristics of the deeper regional division of labor among cities by analyzing the Value Chain Division Index (VCDI). Subsequently, we used the fractional response regression model to analyze influencing factors. We found that, firstly, the core cities are dominated by the production of high-value parts, while the peripheral cities are mainly dominated by the production of middle- and low-value parts. Specifically, northern Anhui, northern Jiangsu, and southwestern Zhejiang are obviously in a disadvantaged position regarding the regional division of labor in the value chain. In the production of middle- and high-value parts, there are close investment connections between the core cities, and only a few peripheral cities maintain a certain degree of connection with the core cities. Therefore, there is a need to further strengthen industrial investment connections between the core and peripheral cities. Secondly, the regional division of labor in the value chain in the Yangtze River Delta region shows the following characteristic: a "one super, many strong" pattern. That is to say, the VCDI value of Shanghai is the highest, and the VCDI value of Suzhou, Ningbo, and Wuxi is also relatively high, while the VCDI value of peripheral cities is relatively low. Furthermore, we found that there is a relatively obvious regional division of labor among cities, but the core cities have strong homogeneity in the high-value areas. Therefore, it is necessary to further strengthen the dislocation of competition between core cities. Thirdly, the model results show that rising land prices and construction in the development zones at the provincial and national levels both have significant contributing effects on the enhancement of the regional division of labor in the value chain, while the innovation inputs, innovation outputs, and their interaction terms show a negative effect. There is a need to further enhance the efficiency of innovation transformation and improve the quality of innovation transformation in order to promote upgrading in the value chain.

**Keywords:** listed enterprises; investment connections; Value Chain Division Index; electronic information manufacturing industry; Yangtze River Delta region





## 1. Introduction

A value chain mainly explains the differences in value acquisition due to the differences in production processes performed by enterprises [1]. Mapping this process to spatial scales, it is manifested as the relative difference in the distribution of regional value acquisition [2]. The main reason for this is that the organization of different production activities by enterprises, according to location factors, leads to the formation of an interconnected regional value chain network [3]. This further leads to the spatial distribution of industrial value chains with a difference in value acquisition, which leads to differences in the spatial distribution of the industrial chain compared to the spatial differences in the distribution of the value chain [4]. This feature is most prominent in the electronic information manufacturing industry [5]. The electronic information manufacturing industry, viewed as a strategic sector reflective of a country or region's scientific and technological competitiveness, receives notable attention from major countries and regions worldwide [6]. The diverse roles played by different countries are due to variances in their scientific and technological bases and research capabilities. The result is that developed countries produce high-value parts, while developing countries are mainly responsible for the production of low-value parts, which creates a differentiated division of labor system on a global scale [7–10].

When this scale turns from global to national, a strong regional division of labor can be found as well [11]. Some cities can play a role in multiple segments of a product's production, while others play a role in a single segment, which creates a larger-scale pattern of regional division of labor in the value chain [12]. On a regional scale, the specialization of urban functions has been continuously shaped and reshaped by the spatial division of labor systems in value chains. Furthermore, it presents the characteristics of a hierarchical difference in the regional division of labor among cities in the value chain [13]. At the same time, the promotion of inter-regional value chain-based division of labor is important for enhancing the competitiveness of regional industries and participating in global competition on behalf of the country.

However, existing urban geography and economic geography studies are relatively insufficient to study the regional division of labor in value chains, and existing studies of industrial value chains have overlooked the role of geospatiality in value chains [14]. In particular, for polycentric global urban agglomeration, the regional division of labor in value chains among cities is not simply a sorted differentiation according to the city hierarchy; instead, it may have more complex characteristics and influencing mechanisms. Therefore, obtaining a comprehensive and systematic understanding of the regional division of labor in the value chains of urban agglomerations and the factors that influence it is still an issue that needs to be addressed urgently by academics.

## 2. Literature Review

Much research has been performed regarding the regional division of labor. In this context, the previous literature primarily approaches this issue from two directions.

Firstly, it extends from industrial value chain analysis to a global value chains perspective, exploring regional divisions of labor [15,16]. The global value chains emerge from the differing value distributions each country attains within the industry or product value chain [17–19]. Under the production organization of global value chains, a governance system dominated by developed countries and subordinated by developing countries has formed [20]. Its main feature is that developed countries produce high-value parts and reap huge profits, while developing countries may be locked into the low-end links of the value chain [21]. For instance, in the electronic information manufacturing industry, pivotal links, like standards, brands, and core components, predominantly reside in the United States and some Western European countries [22]. Simultaneously, Japan, South Korea, Singapore, and Taiwan occupy the middle or high ends of certain value chains, whereas most developing countries find themselves at the low ends, thus establishing a global-scale division pattern of labor in value chains [23–25]. As another example, in the production of cell phone components, although China has a number of cell phone brands,

the production of many key components relies on foreign technology and industry standards [23]. In particular, China is still lagging behind in some high-value areas, such as chip technology and key devices [26]. Although Lee argued that the production of the cell phone components has already formed a production network that crosses national and regional borders [27], core technology components are still mainly distributed in major developed countries around the world. However, transitioning from a global to an intra-country or intra-regional lens reveals regional labor divisions as well [2,3,28]. In the traditional cell phone era, China's cell phone industry was mainly concentrated in a small number of cities, such as Beijing, Tianjin, and Shenzhen [29]. Further, Kang found that there is also a regional division of labor in the production of cell phone parts, with different cities focusing on the different parts of the value chain [30], thus forming a regional production network to a certain extent and constructing a pattern of regional division of labor in the value chain in China [30]. This phenomenon arises from the divergent location preferences for various value chain production segments among cities, subsequently instigating a regional spatial labor division in the value chain [2,13,28]. Yet, owing to data scarcity, studies on regional labor division in value chains remain limited, with even fewer delving into the details of the production of different parts [31].

Secondly, research on regional labor division is conducted via firm network linkages, with the findings extrapolated to city networks. This methodological approach spans from global to national, regional, and inter-city scales. For instance, the GaWC analyzes network linkages among global cities, primarily through the parent–subsidiary connections of advanced productive services firms, such as banking, insurance, law, consulting management, advertising, and accounting, conducting city rankings [32–34]. When these network studies shift to within countries, a regional division of labor can be found under the network of firm connections [2]. The reason for this is that the division of labor in cities has undergone a transformation from inter-industry to inter-sectoral intra-industry or intra-sectoral or intra-product division of labor, and this process has affected the spatial division of labor among cities. Under the effect of industrial agglomeration and diffusion, a differentiated industrial agglomeration pattern of "center–periphery" will gradually be formed [13]. Core cities mainly focus on the service industry, while the peripheral general cities mainly focus on manufacturing [3]. Further, core cities mainly undertake the functions of R&D and design, while the peripheral general cities mainly undertake general manufacturing or assembly [35]. It is very clear that these studies mentioned above were conducted mainly from the perspective of the intra-industry division of labor. However, they typically fail to penetrate the value chain differences in product within firm networks and the characteristics of value chain linkages formed on that basis [36–38]. For example, Wu used the internal enterprise connections of the electronic information industry to look at the characteristics and the basic division of labor of China's urban network [39], but he did not go into the regional division of labor in the value chain. Therefore, the research in this field needs to be strengthened.

Summarizing existing research, these studies have the following major shortcomings. First, most studies have focused on the country scale in the context of global value chains, while there has been a marked lack of studies at the regional scale, which is important for the upgrading of regional value chains and the synergistic development of industries. Second, studies focusing on the division of labor in the firm network have neglected the intra-industry product differences, which are precisely the basis for the urban specialization in the division of labor and require in-depth attention. Building upon these foundations, this paper aims to advance research on regional labor division in value chains based on three aspects. Firstly, it explores the basic characteristics of regional labor division, focusing on the value chain differences in component production by listed enterprises. Secondly, from the perspective of the firm network, it examines the characteristics of city connections in different links of the value chain. This network not only underscores the differences in production activities across the value chain, but also confers geospatial characteristics, bridging industrial economic analysis with the study of urban functional

division of labor [14]. Finally, it proposes a value chain regional labor division index, predicated on the number of enterprises and total number of investment connections in cities in different parts of the value chain. We believe that examining the regional divisions of labor in value chains should account for both of the spatial difference in the production of components and the regional connections presented by the firm network. Only then can we accurately delineate patterns of regional industrial labor division and foster synergistic regional industrial development.

This paper mainly chooses the Yangtze River Delta region as a case. The Yangtze River Delta (YRD) region, deeply engaged in global production and labor division within the electronic information manufacturing field, is a crucial region for China to participate in the competition of global value chains. Despite strong internal city connections, insufficient industrial integration prevents the YRD from gaining significant influence within the global value chain [40,41]. To elevate the YRD region as a global value chain leader, it is imperative to foster an industrial cooperative development system based on labor division in the industrial value chain. Furthermore, deepening the regional division of labor in the value chain will enhance the economic development and improve overall welfare of all of the cities. Simultaneously, it has significant strategic value in promoting Chinese-style modernization and shared prosperity within the YRD.

The structure of this paper is organized as follows. Firstly, it shows the spatial distribution of the value chain of the electronic information manufacturing industry using the distribution of listed companies and firm investment connections in the YRD region. Subsequently, the VCDI is used to demonstrate the characteristics of the regional division of labor in the value chain in the YRD region and to analyze the main development advantages of different cities. Next, we analyze the main influencing factors on the division of labor of the value chain. Finally, some conclusions and discussion for responses are drawn.

## 3. Data Sources and Research Methodology

### 3.1. Data Sources

We utilized the WAND database, the iFinD database, and the Prospective Listed Companies Database (https://stock.qianzhan.com, accessed on 15 May 2022) to establish a 4-digit code database of listed companies in the electronic information manufacturing industry in the Yangtze River Delta region in 2020, specifically including the special equipment manufacturing industry (3562—semiconductor device special equipment), the electrical machinery and equipment manufacturing industry (3841—lithium battery), the computer, communication, and other electronic equipment manufacturing industry, and the electronic information service industry (6520—IC design). Due to it being difficult to measure the value of 5G base stations, fiber optic cables, signal transmission equipment related to communication system equipment, as well as radio and television receiving equipment, radar and ancillary equipment, video surveillance equipment, automotive electronics, and other industries involved in a wide range of value content standards, these industries were not included. Therefore, this paper mainly focused on the consumer electronics industry, such as computer parts, cell phone parts, OEM, and brand enterprises. Eventually, we selected 160 sample listed enterprises, mainly distributed in Shanghai, Suzhou, Hangzhou, and 22 other cities, accounting for about 53% of all cities in the YRD region, which also indicates that nearly half of the cities do not have listed companies in the electronic information manufacturing industry. Meanwhile, we also obtained the distribution addresses and number of investment connections of the subsidiaries and established a parent–subsidiary firm investment connection network. In addition, the city attribute data involved in this paper mainly came from the fourth economic census bulletin, the seventh population census, and the city statistical yearbook in China. The data on land prices came from the website of "Choose Where" (www.xuanzhi.com, accessed on 24 June 2022).

### 3.2. The Classification of Value Chain

Referring to the existing research on value division standards and related research reports on computer parts, cell phone parts, and other consumer electronics parts, while taking into account some components of the technical specificities and specializations, according to the main business type of each listed enterprise, the electronic information manufacturing industry was divided into high-value parts, middle-value parts, and low-value parts [6,23,42,43]. Among them, high-value parts mainly include brands, IC, liquid crystal displays, and optical devices. These components' input thresholds and production costs are high, with strong technical standards and requirements. Middle-value parts mainly include batteries, connectors, PCBs, and many other parts of electronic information products, with relatively high technical requirements. The production of low-value parts mainly involves the production of electronic components to provide a variety of materials, general structural parts, labels, assembly foundries, etc. All these parts have a lower technological threshold and lower cost, belonging to the typical labor-intensive industry. The specific classification results are shown in Table 1.

**Table 1.** Value chain segmentation of electronic information manufacturing industry.

| Classification | Parts Name |
|---|---|
| High-value parts | Electronic brand owners, IC design (analog signal chips, Wi-Fi chips, memory chips, etc.), IC manufacturing (IC foundry, design + manufacturing, etc.), IC packaging, IC materials (polishing fluid, photoresist, silicon wafers, conductive adhesives, repellent materials, electronic special gas), IC test equipment, display device manufacturing (liquid crystal display modules, light guide plates, etc.), optical devices (camera modules, optical filters, etc.) |
| Middle-value parts | Battery components, connectors, PCBs, filters, resonators, oscillators, capacitor assemblies, precision motors, shielding devices, acoustic devices, precision welding and cutting equipment, etc. |
| Low-value parts | Structural components, PCB materials, optoelectronic materials, heat dissipation materials, computer printing materials, labels, cell phone solutions, OEM, etc. |

### 3.3. Value Chain Division Index

The position of cities in the regional division of labor in the value chain is mainly reflected in two aspects: firstly, the overall position of local listed enterprises in the value chain, which intuitively reflects the position of cities in the value chain. This mainly depends on the city's own strength in the development of the electronic information manufacturing industry. Secondly, it is reflected through the establishment of industrial linkages with other cities, not only to provide a channel for cities without local listed enterprises to be embedded in the division of labor in the value chain but also to strengthen the connections with core cities. There is not yet a good index for measuring the level of regional division of labor in the value chain on the city scale. Drawing on the idea of Ge [44] in constructing a global value chain synthesis index, this paper constructs an index based on the number of listed enterprises in different-value segments of a city and the total amount of inter-city mutual investment of listed enterprises and finally creates the Value Chain Division Index (*VCDI*). The $VCDI_i$ is calculated as follows.

$$VCDIF_i = ln\left(1 + \frac{MHVF_i}{TVF}\right) - ln\left(1 + \frac{LVF_i}{TVF}\right) \tag{1}$$

$$VCDIV_i = ln\left(1 + \frac{MHVV_i}{TVV}\right) - ln\left(1 + \frac{LVV_i}{TVV}\right) \tag{2}$$

where $VCDI_i$ denotes the index of regional division of labor in the value chain of city *i* based on the number of listed enterprises; $MHVF_i$ denotes the number of listed enterprises producing middle- and high-value parts of city *i*; $LVF_i$ denotes the number of listed enterprises producing low-value parts in city *i*; *TVF* denotes the number of all listed

enterprises in the YRD region; similarly $VCDIV_i$ denotes the index of the regional division of labor in the value chain of city i based on the investment amount; $MHVV_i$ denotes the total amount of outward investment and investment received by firms in city *i* in the middle- and high-value parts; $LVV_i$ denotes the total amount of outward investment and acquired investment of city *i* in the production of low-value parts; and $TVV$ denotes the sum of outward investment and investment received by listed enterprises in all cities.

Then, the index of the regional division of labor in the value chain of city *i* in the whole region is synthetically expressed by the following equation:

$$VCDI_i = VCDIF_i \times VCDIV_i \tag{3}$$

where $VCDI_i$ denotes the index of the regional division of labor in the value chain of city *i*. Since $VCDIF_i$ and $VCDIV_i$ both have positive and negative values in the value range, in order to avoid the problem of confusing multiplication results due to sign crossing, before synthesizing the $VCDI_i$, a unified standardization method (min–max normalization) was used to compare the value chain of $VCDIF_i$ and $VCDIV_i$ standardization processing. The min–max normalization is a linear variation of the original data that maps the values to between [0, 1]. When $VCDI_i$ is close to 1, it indicates that city *i* has a comparative advantage in the production of high-value parts; when converging to 0, it indicates that the city has a comparative advantage in the production of low-value parts.

### 3.4. Fractional Response Regression Model

Since the explanatory variables belong to the exponential values from 0 to 1, the fractional response regression model was used for estimation. The reasons for using this model are as follows. First, inaccuracies in the model description and statistical results can be avoided. Second, if an ordinary regression model is used, the predictions may go beyond these intervals. Third, the model is effective in capturing specific nonlinear relationships when the outcome variable is close to 0 or 1 [45,46]. The specific formulas are as follows:

$$VCDI_i = \Phi(\beta_0 + X\beta) = \Phi(\beta_0 + \beta_1 x_1 + \ldots + \beta_k x_k + \omega_t) \tag{4}$$

where $VCDI_i$ is the index of the regional division of labor in the value chain of city *i*. $\Phi(\cdot)$ is the normal distribution function, since the value of $\Phi(\cdot)$ always lies between 0 and 1, thus ensuring that the dependent variable of $VCDI_i$ always lies between 0 and 1.

### 4. Characteristics of Regional Division of Labor in Value Chain

### 4.1. Spatial Differentiation of Value Chain

Based on the number of listed enterprises in each link of the value chain in the electronic information manufacturing industry, using the spatial classification method of ArcGIS 10.2 software to visualize them, the results are shown in Figure 1. There exists a more obvious division of labor of the value chain within the YRD region, with the high-value parts mainly being produced in the core cities, which occupy the leading position. Meanwhile, vast areas of northern Anhui, southern Anhui, northern Jiangsu, and southwestern Zhejiang mainly produce middle- or low-value parts, meaning they are obviously in a disadvantaged position in the regional division of labor. Specifically, high-value parts are mainly produced in Shanghai, Suzhou, and Wuxi, with 31, 20, and 9 listed enterprises, respectively, with a strong quantitative distribution advantage. At the same time, Hangzhou, Hefei, Changzhou, Ningbo, Nanjing, and other core cities have four or five listed enterprises producing high-value parts, having a certain advantage. Nantong, Zhenjiang, Jiaxing, and other cities, although they have fewer listed enterprises that cut into this link, still have two or three listed enterprises. In addition, in the periphery, Suqian, Tongling, Bengbu, and other cities also have a listed enterprise producing high-value parts. The distribution of enterprises producing middle-value parts is mainly centered on Suzhou, with 10 listed enterprises. Changzhou, Shanghai, Ningbo, and Wenzhou are the sub-centers, all with three to five listed enterprises. Low-value parts are mainly produced in Suzhou,

Shanghai, and Hangzhou, which have nine, four, and two listed enterprises, respectively. It can be seen that the listed enterprises producing low-value parts still tend to gather in the core cities. Assembly and foundry enterprises are concentrated in Shanghai and Jiaxing and have a strong competitive advantage in the region. Finally, Ningbo Bird Company, located in Ningbo, is the only local terminal brand enterprise. However, compared with Huawei, OPPO, Vivo, and other global brands, the competitiveness is very weak.

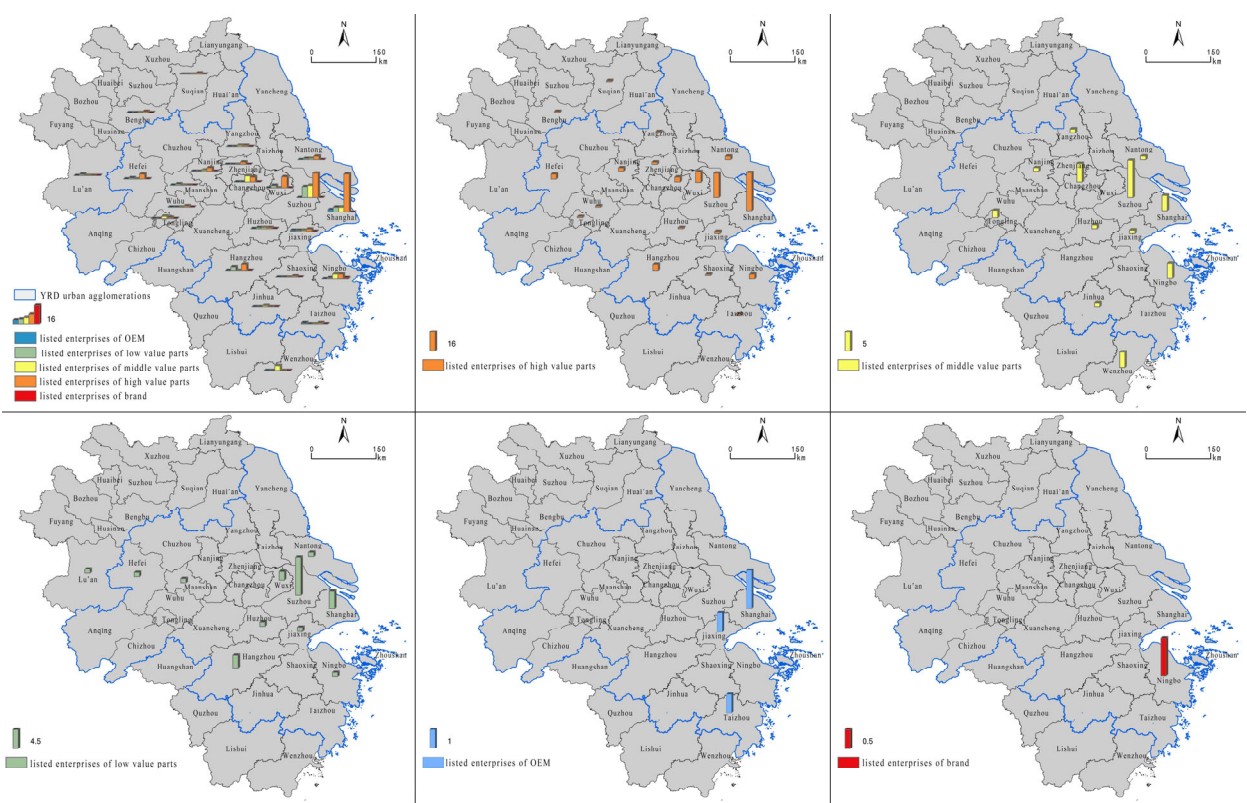

**Figure 1.** Spatial distribution of listed enterprises manufacturing parts of differing value in the YRD region in 2020.

### 4.2. Networking Connections of Value Chain

Through the previous analysis, it is found that cities with listed enterprises in the electronic information manufacturing industry only account for half of the total number of cities in the YRD region, indicating that a large number of cities are still excluded. Therefore, the investment network comprising parent–subsidiary connections of listed enterprises is used to further delineate the city connections between the production of parts of differing value. At the same time, in order to examine whether there are other cities that are integrated into the regional division of labor in the value chain by attracting listed enterprises, we use the natural break method in ArcGIS 10.2 software to classify the number of investment connections, and the results are shown in Figure 2. From Figure 2, we can see that the core cities are closely connected with each other, while the peripheral cities are also embedded in the regional industrial system, but the connections with the core cities are still weak. Among them, in the production of middle- and high-value parts, the core cities are closely connected with each other, such as Shanghai, Suzhou, Wuxi, Hefei, and Ningbo, which are very closely connected, indicating that enterprises producing middle- and high-value parts tend to invest in cities of the same level. Specifically, there are close connections between core cities in the production of middle- and high-value parts, such as the total investment connections between Shanghai and Shaoxing, Wuxi and Suzhou, Shanghai and Ningbo, Hefei and Suzhou, and Ningbo and Nanjing amounting to CNY 15, 10.55, 5.86, 4.03, and 3.83 billion, respectively. Among them, Shanghai's outward investment intensity

in the production of middle- and high-value parts is high. The cities that mainly obtained Shanghai's outward investment are Shaoxing and Ningbo. The investment in Shaoxing is dominated by two listed enterprises, Will Semiconductor and SMIC. At the same time, SMIC's investment in Ningbo is near to CNY 5 billion. Furthermore, Suzhou and Wuxi are also very closely interconnected. This is mainly because JCET's investment in Suzhou is absolutely dominant, with an investment scale of close to CNY 3 billion, while Suzhou's investment in Wuxi is relatively small. At the same time, some cities in the periphery region participate in the regional division of labor in the value chain by attracting the investment of listed enterprises producing middle- or high-value parts. For example, Suqian makes up for its absence in the regional division of labor in the value chain by attracting investment from JCET. Quzhou makes up for its absence by attracting investment from Hangzhou Lion Micro. However, most peripheral cities are still relatively weakly connected to the core cities.

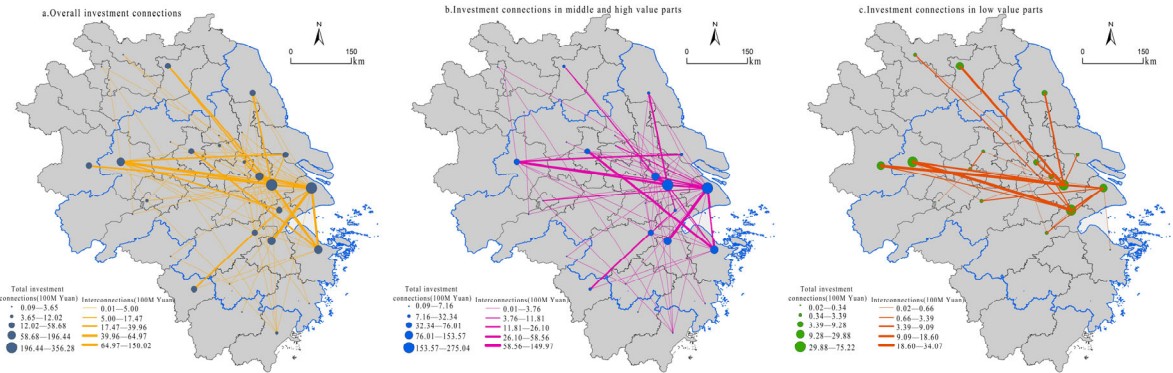

**Figure 2.** Investment connections network of the value chain in the YRD region in 2020.

In the production of low-value parts, the internal connections within the YRD region are dominated by the core cities, with cities in the north of Jiangsu Province and Anhui Province. However, there are basically no connections between cores cities and cities in Zhejiang Province. For example, Suzhou–Lu'an, Suzhou–Suqian, Suzhou–Yancheng, and other core cities have generated close connections with peripheral cities, and Yancheng and Lu'an, in particular, are deeply involved in the entire network through their close connections with Suzhou. Furthermore, Yancheng has attracted investment from Suzhou Cosun Technology and Dongshan Precision, which has contributed to the rapid development of the local electronic information manufacturing industry. The electronic information manufacturing industry accounted for 6.4% of Yancheng's manufacturing output in 2020, making it one of the leading industries in the region. (Source of data: The Statistical Yearbook of Yancheng). Lu'an has promoted the rapid development of its electronic information manufacturing industry by attracting the investment of two major structural component enterprises, Victory Precision and Chunxing Precision from Suzhou, and has made electronic structural components one of the main focuses of its future new-generation information technology industry. In addition, there is a relatively close connection between Jiaxing and Xuzhou, indicating that peripheral cities can also be embedded in the regional division of labor system by establishing industrial connections with core cities.

*4.3. Characteristics of Regional Division of Labor in Value Chain*

On the basis of the previous analysis, the VCDI in different cities is calculated to further reveal the position of each city in the regional division pattern of labor, and then we use the natural break method in ArcGIS 10.2 software to classify the VCDI, and the results are shown in Figure 3. From Hangzhou–Nanjing–Hefei to the north of Jiangsu Province and then to the southwest of Zhejiang Province and Anhui Province, the VCDI shows a trend of gradual decline. This result is also consistent with the general perception that core cities are usually engaged in the production of middle- and high-value parts, while peripheral

cities are mainly engaged in the production of low-value parts, showing a certain regional spatial division of labor. Meanwhile, it displays a "one super, many strong" pattern within the core cities, with Shanghai as the center and Suzhou, Ningbo, Wuxi, and other cities as sub-centers, confirming to a certain extent the polycentric character. Specifically, Shanghai's VCDI is the highest at 1, with a prominent advantage in the production of middle- and high-value parts. Suzhou's VCDI is the second highest at 0.44, with an obvious advantage in the production of middle- and high-value parts. Ningbo and Wuxi also have a stronger advantage in the production of middle- and high-value parts, with a VCDI of 0.18 and 0.17, respectively. Meanwhile, Nanjing, Hefei, Hangzhou, and other cities have some relative advantages in the production of middle- and high-value parts, but the VCDI is between 0.02 and 0.05, which is still relatively low compared to Shanghai and Suzhou. The VCDIs in the northern part of Jiangsu Province, the northern and southern part of Anhui Province, and the vast area of the southwestern region of Zhejiang Province are significantly lower than 0.01, and there is a large gap between them and the major cities. Therefore, this also shows that these areas are in a disadvantageous position in the regional division of labor in the value chain.

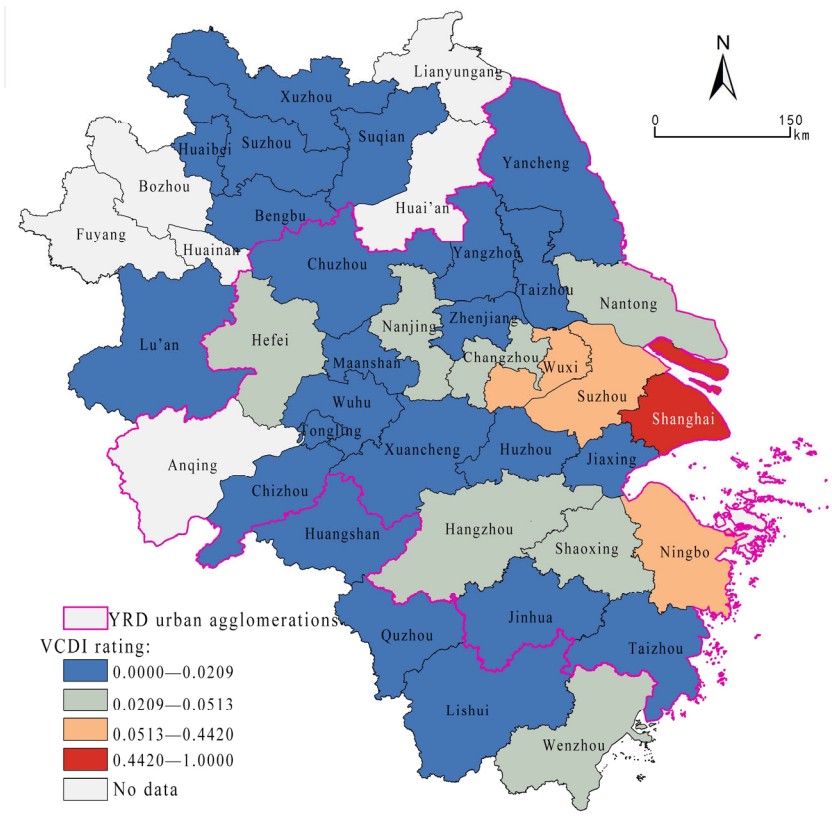

**Figure 3.** Spatial differentiation of the VCDI in the YRD region.

The deeper features of the regional division of labor within the Yangtze River Delta region are further revealed by comparing the product differentiation and main features of each city in the production of parts of differing value, and the results are shown in Table 2. After comparing the main products of listed enterprises in major cities, it is found that there is a certain division of labor of the value chain within the core cities in the YRD region, but the competition is more intense in the production of middle- and high-value parts. Shanghai and Suzhou are the two major cities with the strongest advantage in the production of high-value parts. In particular, Shanghai dominates the production of IC manufacturing parts, while Suzhou has outstanding advantages in the fields of liquid crystal displays, precision structural components, etc. Additionally, a number of enterprises, such as Anjie Technology, Victory Precision, and Dongshan Precision, have cut into the supply chains of

Apple and Samsung. Ningbo and Wuxi have also obtained a clear advantage in the regional division of labor in the value chain. Wuxi has certain advantages in the field of IC design, packaging, materials, etc. Ningbo is mainly involved in the field of optical devices and battery components, displaying development advantages, and also has a brand enterprise waveguide share and is dominant in the production of middle- and high-value parts and brands. Hangzhou, Nanjing, and Hefei have certain advantages in the production of ICs, liquid crystal displays, and other high-value parts, although the VCDI is low. Nantong and Changzhou are also developing in niche areas due to their proximity to the core cities, both of which are making an effort in the production of IC parts. Wenzhou, Yangzhou, Shaoxing, and other cities are mainly involved in certain links in the value chain to participate in the regional division of labor. Finally, although Tongling is a small city, it is involved in multiple stages of the production of middle- and high-value parts.

**Table 2.** Comparison of product types and main characteristics of major cities in the YRD in 2020.

| City | Product Type | Key Features |
| --- | --- | --- |
| Shanghai | IC design, IC manufacturing, IC packaging and testing, plating, cleaning equipment, semiconductor materials, liquid crystal displays, connectors, communication modules, structural components, PCB materials, communication peripherals, cell phone solution design, and many other parts | Focus on high-value parts, especially IC industry chain is perfect, and control OEM |
| Suzhou | Liquid crystal displays, IC design, IC packaging, IC testing equipment, semiconductor materials, PCB, connectors, filters, functional protective devices, precision structural components, and many other parts | Focus on medium- and high-value parts, with advantages in liquid crystal display and precision structural components |
| Ningbo | Camera modules, backlight modules, optical modules, semiconductor materials, battery components, connectors, cell phone brands, and many other fields | Focus on middle- and high-value parts, with advantages in optics, batteries, and brands |
| Wuxi | IC design, IC packaging, semiconductor materials, heat dissipation materials, structural components | Focus on IC design and packaging |
| Hangzhou | IC design, IC special equipment, semiconductor materials, optical devices, PCB materials, chassis, and many other fields | Focus on high-value parts, with certain strengths in the IC design |
| Nanjing | Liquid crystal displays, PCB, etc. | Focus on high-value parts, LCDs |
| Hefei | Specialized equipment for ICs, communication security equipment, liquid crystal displays, magnetic materials | Focus on multiple areas of high-value parts |
| Nantong | IC design, IC packaging, aluminum capacitors, aluminum electronic materials | Focus on IC design and packaging |
| Wenzhou | Precision electronic connectors | Focus on medium-value parts |
| Changzhou | IC design, IC packaging, IC equipment, semiconductor materials, PCB, etc. | Has advantage in IC and PCB |
| Yangzhou | IC manufacturing + packaging, PCB | Focus on high-value parts |
| Shaoxing | IC packaging and testing equipment | Focus on high-value parts |
| Zhenjiang | IC packaging, liquid crystal displays | Focus on high-value parts |
| Tongling | IC packaging and testing equipment, crystals, capacitors | Access to multiple areas of middle- and high-value parts |

Note: We only list cities with high VCDI and listed enterprises; the remaining cities are not included.

## 5. Driver Factors Analysis

### 5.1. Description of Variables

In terms of the selection of variables, referring to the existing research results [47–50], it is mainly based on seven aspects, industrial base, innovation capacity, economic strength, international connection, production cost, infrastructure, and government involvement.

Firstly, a good industrial foundation can provide specialized labor, shared intermediate inputs, and technological overflow for the development of local electronic information manufacturing industry. Further, it can attract the investment of electronic information manufacturing enterprises, and then promote the development of electronic information

manufacturing industry through industrial agglomeration. At the same time, with the continuous consolidation of the industrial foundation, the electronic information manufacturing industry will continue to mature; market competition is also constantly strengthened, and only businesses in the value chain of high-end industries can survive in the face of such fierce competition, thus encouraging the city to enhance its position in the regional division of labor in the value chain. Therefore, the foundation of industrial development is mainly portrayed by two indicators, namely the number of employees in the electronic information manufacturing industry and the business income of the electronic information manufacturing industry. Secondly, innovation ability is the core element of the city's ability to begin to produce high-value parts. Only continuous innovation input and innovation output can control the core parts. Therefore, we choose innovation input (R&D input accounting for the proportion of operating income) and innovation output (number of patents granted) as two indicators to reflect the level of the city's innovation input and innovation output. Thirdly, economic strength can provide solid economic support for the city's industrial development and encourage cities to produce high-value parts. At the same time, enterprises are more inclined to invest in cities with better economic foundations, especially enterprises producing high-value parts. Therefore, economic strength is mainly expressed by two indicators, the per capita gross domestic product and urbanization level. Fourthly, by connecting with foreign markets, it is possible to establish local industrial connections with the rest of the world. A city's active participation in globalization is an important way for it to access foreign technological spillovers and expand its market, which will have a significant impact on the position in the value chain. Therefore, international linkages are mainly expressed by two variables: one is the total imports and exports and the other is foreign direct investment. Fifthly, a city can attract the investment of enterprises belonging to different segments of the value chain through the comparative advantage of local production costs. Therefore, labor wages and land prices are chosen to indicate this. Sixthly, good infrastructure, especially convenient transportation conditions, is more likely to attract enterprises belonging to different segments of the value chain. So, the per capita urban road area is chosen to indicate the level of infrastructure development. Finally, through strong governmental support in the form of finance and taxation, local electronic information manufacturing clusters are cultivated, and foreign enterprises are encouraged to invest, gradually enhancing these cities' position in the regional division of labor in the value chain. Therefore, government involvement is mainly chosen to be represented by general fiscal budget expenditure and the number of electronic information manufacturing development zones. All variables are shown in Table 3.

**Table 3.** Attribute characteristics of variables in the model.

| Characterization | Variable | Description | Maximum | Minimum | Average | Median | Standard Deviation |
|---|---|---|---|---|---|---|---|
| Industrial foundation | Ln(Employment) | Number of employees | 4.617 | −1.801 | 1.069 | 1.284 | 1.466 |
| | Ln(Revenue) | Operating income | 9.273 | 2.120 | 5.38 | 5.407 | 1.74 |
| Innovation ability | R&D | R&D investment as a percentage of revenue | 4.790 | 0.270 | 2.404 | 2.005 | 1.145 |
| | Ln(Patent) | Patent grants | 11.848 | 7.345 | 9.792 | 9.949 | 1.195 |
| Economic strength | Ln(PGDP) | GDP per capita | 12.019 | 10.543 | 11.417 | 11.431 | 0.391 |
| | Urbanization level | Urbanization rate | 89.3 | 43.76 | 68.654 | 68.125 | 10.41 |
| International connections | Ln(Im-export) | Total import and export trade | 10.459 | 4.245 | 6.851 | 6.818 | 1.651 |
| | Ln(FDI) | FDI | 6.208 | 0.884 | 2.963 | 2.947 | 1.292 |
| Production costs | Ln(Labor) | Average wage of urban on-the-job workers | 12.055 | 11.180 | 11.462 | 11.411 | 0.187 |
| | Ln(Land) | "Tendering, auctioning and listing" land prices | 4.377 | 2.083 | 3.083 | 3.023 | 0.627 |

**Table 3.** *Cont.*

| Characterization | Variable | Description | Maximum | Minimum | Average | Median | Standard Deviation |
|---|---|---|---|---|---|---|---|
| Infrastructures | Ln(Road) | Urban road area per capita | 3.049 | 1.596 | 2.301 | 2.327 | 0.342 |
| | Ln(Expenditures) | General budget expenditure | 9.000 | 5.182 | 6.495 | 6.472 | 0.793 |
| Government involvement | Ln(Development zone) | Number of electronic information manufacturing development zones above the provincial level | 3.638 | −18.421 | −0.432 | 1.099 | 5.657 |

### 5.2. Model Regression Results

Before using the fractional response regression model, we first check whether there is collinearity between the variables, and it is found that there is a strong collinearity. Then, by means of the step-by-step test, it is finally found that the collinearity among variables of R&D, urbanization level, Ln(FDI), Ln(Land), Ln(Road), and Ln(Development zone) is the lowest. Therefore, these variables were selected for model regression, and the results of the collinearity test are shown in Table 4.

**Table 4.** Independent variables' collinearity test results.

| Variables | R&D | Urbanization Level | Ln(FDI) | Ln(Land) | Ln(Road) | Ln(Development Zone) |
|---|---|---|---|---|---|---|
| R&D | 1 | | | | | |
| Urbanization level | 0.0783 | 1 | | | | |
| Ln(FDI) | 0.1407 | 0.6456 | 1 | | | |
| Ln(Land) | 0.2619 | 0.6595 | 0.6307 | 1 | | |
| Ln(Road) | −0.3095 | 0.2975 | 0.1130 | −0.1350 | 1 | |
| Ln(Development zone) | −0.0771 | 0.2881 | 0.0563 | −0.1971 | 0.0321 | 1 |

Using Stata 14 software to conduct the fractional response regression model to test the impact of variables such as the R&D, urbanization level, FDI, and other variables, and the results are shown in Table 5. Model 1 shows that R&D is significantly negative, which is inconsistent with the conclusions of previous studies and general cognition. Then, the variable is replaced by the innovation output, and then regressed using the fractional response regression model with other variables unchanged, and the results are shown in the Model 2 column of Table 5. On this basis, the interaction terms of innovation input and innovation output were established, and the regression results are shown in the Model 3 column of Table 5.

From the regression results of Model 1 in Table 5, the results of R&D are significantly negative. This indicates that the improvement of the level of innovation input does not significantly enhance the city's position in the regional division of labor in the value chain. This is a big difference from previous research, and for this reason, the innovation input indicator is replaced by the innovation output indicator. This is because research found that the higher the number of patents granted, the better the city's innovation ability. So, the enhancement of the innovation ability can also promote the development of the city's electronic information manufacturing industry, improving its technological level and competitiveness. Furthermore, it will enhance the city's position in the regional division of labor in the value chain [47,50]. However, after replacing the patent with R&D, the result of Model 2 finds that it has a positive coefficient, but the regression result is not significant. Further, to consider the innovation capacity of cities, it is necessary to comprehensively examine the innovation input and output, and so the interaction term of innovation input and innovation output is utilized to conduct a re-regression (Model 3), and the result

still finds that this variable is significantly negative. Compared with previous studies, Xiong [49] obtained the same result in the study of the division of labor in the value chain in East Asian countries, which is due to the lower conversion rate of R&D results. At the same time, the expenditure on environmental improvement to optimize the division of labor in the value chain is more limited. Therefore, the increase in R&D investment will crowd out other expenditure on the quality of the system and the hardware infrastructure. Finally, it may not be conducive to the enhancement of the city's position in the regional division of labor in the value chain [49].

**Table 5.** Fractional response regression results.

| Variables | Model 1 | Model 2 | Model 3 |
|---|---|---|---|
| R&D | −0.4614 *** (0.1391) | - | - |
| Ln(Patent) | - | 0.5860 (0.3683) | - |
| R&D × Ln(Patent) | - | - | −0.0449 *** (0.0127) |
| Urbanization level | 0.0012 (0.0339) | −0.0443 (0.0339) | 0.0060 (0.0344) |
| Ln(FDI) | 0.0917 (0.1031) | −0.1165 (0.1856) | 0.1189 (0.1071) |
| Ln(Land) | 1.3681 ** (0.6513) | 1.3533 * (0.7940) | 1.3778 ** (0.6351) |
| Ln(Road) | 0.5276 (0.7134) | 1.5718 * (0.8058) | 0.4555 (0.6901) |
| Ln(Development zone) | 0.0775 * (1.7151) | 0.1267 (0.0979) | 0.0763 ** (0.0390) |
| Constant | −6.8981 *** (11.8641) | −12.4190 (2.9985) | −7.1854 *** (1.6513) |
| Samples | 34 | 34 | 34 |
| Pseudo R2 | 0.4530 | 0.4093 | 0.4569 |

Note: Robust standard error in parentheses, *** $p < 0.01$; ** $p < 0.05$; * $p < 0.1$.

From the variables of production cost, the land variable shows a significant positive effect, indicating that a higher price of land is conducive to the improvement of the city's position in the regional division of labor in the value chain. This is mainly due to the fact that industrial land in the YRD region is relatively scarce and under strict control, resulting in relatively high land costs. Simultaneously, cities with higher land costs are always the core cities in the YRD region, such as Shanghai, Suzhou, Nanjing, Hangzhou, and other cities with higher land costs. In view of the unique advantages of the core cities in terms of human resources, science and technology, capital, market, and so on, the enterprises producing middle- and high-value parts will still be willing to stay there [48]. This also shows that the regulation of land prices can promote changes in the location choices of electronic information manufacturing enterprises producing parts of differing value. Enterprises producing low-value parts will leave the local area because they cannot afford the high land costs, while enterprises producing middle- and high-value parts are still willing to stay in the core cities, which is highly dependent on the advantageous elements of the core cities. Due to core cities having strong economic benefits and market competitiveness, this promotes the city's position in the value chain in the form of reverse coercion. Then, the city's position in the regional division of labor in the value chain will be upgraded.

Finally, the regression result of the variable of development zones is significantly positive, indicating that the construction of electronic information manufacturing development zones has a positive role in promoting the city's position in the regional division of labor in the value chain. The reason for this is that the higher the hierarchy of the development zone, the more policy support and incentives can be obtained from the government and the

more value that can be created and captured by the electronic information manufacturing enterprises [50]. At the same time, development zones are conducive to the formation of an agglomeration economy, which is conducive to encouraging enterprises to strengthen technological connections and innovation overflow. Then, it will promote the technological change and functional upgrading of the city in the value chain, and then enhance the city's position in the regional division of labor in the value chain. For example, Shanghai has two national high-tech zones, the Zhangjiang High-tech Industrial Development Zone and the Caohejing Emerging Technology Development Zone. Meanwhile, the electronic information industry is one of its leading industries, which encourages a lot of enterprises producing high-value parts to relocate to development zones. Suzhou has formed an electronic manufacturing industry cluster relying on the Suzhou Industrial Park, such as the Suzhou High-tech Development Zone and other national development zones. Similarly, Wuxi and Ningbo have also formed electronic manufacturing industry clusters. Consequently, Wuxi and Ningbo's electronics and information enterprises producing high-value parts are mainly located in their development zones. To sum up, the construction of high-grade economic and technological development zones with electronic information manufacturing as the dominant industry will be conducive to the city's position in the regional division of labor in the value chain.

## 6. Conclusions and Discussion

### 6.1. Conclusions

This paper analyzes the regional division pattern of labor and the connection characteristics of the value chain of the electronic information manufacturing industry in the YRD region. Furthermore, based on the number of listed enterprises and the parent–subsidiary investment connections between cities, we build a Value Chain Division Index to portray the position of cities in the regional division of labor in the value chain. Then, this paper analyzes the main influencing factors. Through these analyses, the main conclusions are as follows.

Firstly, the core cities in the YRD region are mainly dominated by listed enterprises producing high-value parts, while the peripheral cities are mainly dominated by enterprises producing middle- and low-value parts. In particular, the cities in the north and south of Anhui Province, the north of Jiangsu Province, and the southwest of Zhejiang Province are disadvantaged in the regional division of labor in the value chain. This is due to the fact that talents, technologies, and various production resources in the YRD region are mainly concentrated in the core cities, while the peripheral cities are in a backward position in the regional division of labor in the value chain. Similarly, the core cities are closely connected with each other in the production of middle- and high-value parts. Although a small number of peripheral cities are embedded in the regional industrial division of labor by attracting enterprises producing high-value parts, the network connections between core and peripheral cities remain weak. In the production of low-value parts, the cities of Zhejiang Province are weakly connected to other cities in the YRD region, with the core cities mainly being connected to cities in the north of Jiangsu Province and Anhui Province. Peripheral cities such as Suqian, Yancheng, Lu'an, and Quzhou are deeply integrated into the regional industrial division of labor by maintaining close ties with core cities such as Suzhou and Hangzhou.

Secondly, the pattern of the regional division of labor in the value chain within the YRD region shows more significant polycentric characteristics, which also confirms the polycentricity of the urban network in the YRD region. Furthermore, different levels of cities show stronger characteristics of a regional division of labor. For example, Shanghai is at the core position in the whole region, and Suzhou is in second place, showing a pattern of "one super, many strong" for the division of labor in the whole region. Although the VCDI in some core cities is low, they have strong advantages regarding the production of middle- and high-value parts. These cities include Hangzhou, Ningbo, Hefei, Nanjing, etc., all of which have advantages in the fields of ICs, panels, and optical devices. In

addition, Tongling and other peripheral cities are also involved in the division of labor in the manufacture of high-value parts. This shows that the intra-regional division of labor in the value chain system is not simply distributed according to the city hierarchy but shows more complex characteristics and internal connections.

Thirdly, the regression results of innovation input, innovation output, and the interaction terms all find that innovation factors fail to enhance the cities' position in the regional division of labor in the value chain. This indicates that the quality of innovation in the cities needs to be further improved, and it is necessary to pay more attention to the transformation of the application market of innovation factors. Only in this way can innovation ability be transformed into new productivity and production methods. As a result, the cities' position in the regional division of labor in the value chain can be improved. The rising cost of land will enhance the city's position in the regional division of labor in the value chain. This means that with the rapid rise in land prices, enterprises that produce low-value parts will leave the core cities, while enterprises that produce middle- and high-value parts will still tend to concentrate in the core cities. The construction of development zones above the provincial level with the electronic information industry as one of the leading industries will be conducive to the upgrading of cities in the regional division of labor in the value chain, because it promotes the upgrading of the cities in the regional division of labor in the value chain through policy support, tax incentives, agglomeration economy, and technological spillover within the development zones.

*6.2. Discussion*

Although many previous studies have focused on the division of labor from a value chain perspective, most researchers analyzed this question from the perspective of global value chains and paid less attention to the regional scale. In particular, the studies at the urban scale within the region have been neglected. This is mainly due to the lack of data, which makes it difficult to measure the position in the value chain. At the same time, existing studies have overlooked the intra-regional industrial linkages, especially the intra-product specialization and linkages of a particular industry. Based on this, this paper draws on the value of components of the consumer electronics industry and classifies the value of their components, so as to analyze the position of each city in the regional division of labor. At the same time, the investment connections to the subsidiary companies by listed companies are utilized to comprehensively portray the city connections in different segments of the value chain. We also use the number of listed companies and the number of investment connections in the manufacture of parts of differing value to build an index named the VCDI. Furthermore, we use it to analyze the position of different cities in the regional division of labor in the value chain. Compared to some previous studies, we adopt a more scientific approach to measure the position of cities in the regional division of labor, not just based on the types of products which are produced by a city [23,43].

Furthermore, it is the creation of this index that portrays the regional division of labor that allows us to apply the regression model. The results for some of the variables are consistent with existing research, except for the variable of innovation ability. Although the government always emphasizes the role of innovation, at the same time, academic studies have also discussed the role of innovation in promoting regional development [51]. On this point, we also agree to a large extent. However, the regression result is negative, both for the variables of innovation input and innovation output. Such a result seems to be different from the findings of existing studies [50]. However, we still believe that only continuous innovation can enhance the position of cities in the regional division of labor in the value chain. The reason for such regression results, we believe, is mainly because some innovation input or output is not really transformed into productivity. This means that R&D inputs or the numbers of patents are not really transformed into innovation momentum, which can enhance the city's position. This suggests that cities should be very concerned with the transformation of innovation efficiency, make breakthroughs in key technological links, and improve the quality of innovation.

Finally, in the process of region development, we find that the core cities are always in a favorable position and the peripheral cities receive relatively few development opportunities [2]. At the same time, core cities are more likely to connect with each other than with peripheral cities, which is in line with existing studies that have found that developed countries are more inclined to invest in other developed countries [52]. Regional differences between developed and less developed countries may be difficult to resolve. However, there is still a need to promote technological spillovers, industrial transfers, and financial support from core cities to peripheral cities in a region, so as to minimize the intra-regional differences and promote synergistic development of the whole region [53]. This is one of the goals of the Chinese government, and it is actively promoting the synergistic and integrated development of the YRD region [40]. More importantly, core cities need to drive the development of the peripheral cities, thus further narrowing the internal differences of the region and forming a regional growth pole with global influence [40,41,53]. Therefore, although we know that it is difficult to do so, it is still necessary to guide the core cities to invest more in the peripheral cities through the guidance of the government, strengthen the industrial links between the core and peripheral cities, and then drive the economic development of the peripheral cities.

This study has some limitations. Firstly, this paper argues that the study of the regional division of labor in the value chain can be comprehensively portrayed by using the headquarters of listed enterprises belonging to different parts of the value chain as well as mutual investment connections. Due to the difficulty in obtaining more in-depth data on the value distribution of the cities, the grading of the value chain still needs to be further corroborated in the future by means of new data and practical research. At the same time, the index construction of the VCDI mainly draws on the research methods from international trade study, which may not necessarily be very mature and need to be further tested in the future. Secondly, in-depth comparative analysis of multiple industries is also needed to further reveal the pattern and trend of the regional division of labor in the value chain within the YRD region, so as to better propose countermeasures for the synergistic development of regional industries and the high-quality development of this region. For example, studies could select the automobile industry or the biomedical industry to look at the spatial distribution and region connections of different parts of the value chain of these industries and to further reveal the characteristics of the refinement of the intra-regional division of labor. Finally, analyzing the regional division of labor from the perspective of the value chain is a better entry point, because such a study needs to integrate the research of different research fields such as global value chains, the industrial economy, the regional economy, and urban geography, and we suggest that more studies should perform analysis in this field.

**Author Contributions:** Conceptualization, J.K. and Y.N.; investigation, J.K. and C.Y.; resources, J.K.; writing—original draft preparation, J.K.; writing—review and editing, J.K. and Y.N.; supervision, J.K. and C.Y. All authors have read and agreed to the published version of the manuscript.

**Funding:** This research was supported by the National Natural Science Foundation of China (42101213).

**Institutional Review Board Statement:** Not applicable.

**Informed Consent Statement:** Not applicable.

**Data Availability Statement:** The data can be obtained from the corresponding author upon reasonable request.

**Conflicts of Interest:** The authors declare no conflict of interest.

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
