# Peer review of "Analysis of Regional Division of Labor in Value Chain Patterns and Driving Factors in the Yangtze River Delta Region Using the Electronic Information Manufacturing Industry as an Example"

_sustainability, doi:10.3390/su151914393_

Round 1

Reviewer 1 Report

There are several grammar and typos in the paper that should be considered. 

Author Response

1、the introduction section should be divided into two sections, introduction, respectively literature review. The literature review section is very weak, and the research gap is not explained.

Response:

Thanks for the constructive comment. In this paper, the introduction and literature review have been reorganized and described separately, with more content and relevant references added.

2、Table 3 adding information about the median and standard deviation.

Response:

Thanks for the suggestion. The median and standard deviation are added in Table 3

3、Specify the software used to conduct the fractional response regression

Response:

Thanks for the suggestion. We use stata14 software conduct the fractional response regression. In response, we add a sentence in the first paragraph of page 15.

4、Specify the software used to realize the maps.

Response:

Thanks for the suggestion. We provide a methodological description in the beginning of sections 2.1, 2.2 and 2.3 of the paper.

5、In section 3.2, you discuss the covariance test results, while Table 4 presents the results of the collinearity tests. Collinearity is not the same thing as covariance. Address this aspect.

Response:

Thanks for the problem. This paragraph mainly discusses the collinearity of variables, which has been revised in page 15.

6、The discussion section is weak. Several practical and theoretical implications should be discussed.

Response:

Thanks for the suggestion. we delete the section of recommendations, and added the discussion. The discussion part mainly discusses the advanced nature of VCDI compared with existing research methods, the differences between the results of innovation variables and previous studies, and the importance of strengthening the industrial links between core cities and peripheral cities. The revised section can see in page 19 to 20.

7、There were no limitations or future research directions provided

Response:

Thanks for the suggestions. We have added a paragraph at the section of “Conclusion to Discussion” to point out the limitations or future research. See the article for details in the last paragraph of page19

8、There are several grammar and typos in the paper that should be considered.

Response:

Thanks for the suggestion. We double-checked the words, grammar and tenses of this paper and made careful modifications on this.

Reviewer 2 Report

This article aims to analyze the patterns and characteristics of the regional division of labor within the value chains of the electronic information manufacturing industry in the Yangtze River Delta region. By developing a Value Chain Division Index, the authors map the positions of major cities within the regional labor division and identify the advantages of segmented areas. The main contribution of this article is in revealing complex labor division patterns that are not only distributed based on city hierarchy but also show more intricate characteristics and connections.

General Concept Comments:

Abstract

Overall, the abstract has successfully presented most of the important aspects of the research. However, some adjustments must be made to provide a more informative overview.

·   The authors need to add a little information about this study's background or broader context. An explanation of the importance of this topic in a global or national context might enhance clarity and relevance.

·  Discussing the practical implications of these findings could provide additional insights into the importance of this research.

Introduction

Overall, the introduction section is relatively strong and informative, successfully establishing this research's context, relevance, and purpose. However, there need to be some adjustments in terms of clarity, structuring, and transitions, as follows:

·   Although providing much information, this introduction is lengthy and dense. This may overwhelm the reader and can blur the key points. If information is not directly relevant to the research objectives or questions, consider reducing it or moving it to another section of the article.

·         Some sentences may be overly complex or lengthy. They could be divided into shorter sentences without losing meaning.

·  Some sections might require smoother transitions between points, which would assist in guiding the reader through complex arguments. In this case, some sections might feel abrupt or disconnected. Adding transition sentences between different sections can help guide the reader. For example, after explaining the industrial background, you could add, "In this context, previous research has revealed..." to move the reader to the literature review.

·  Ensure that each section follows logically from the one before. If necessary, rearrange some paragraphs or sections to ensure a more logical flow.

·    Check if any sudden changes in style or tone might disturb the reader. Maintain a consistent style and tone throughout all sections.

·  Although the author mentions the importance of this topic in the regional context, adding information about its global and national significance might provide a broader context..

Result and discussion

Overall, this section has provided an in-depth analysis but requires some corrections and additions to enhance the quality and understanding of the reader.

·   Some statements may be confusing or unclear, such as the word "finance," which seems to be repeated in the same context.

·  Several grammatical and syntactical errors can disrupt the flow of reading. Some examples include:

ü  Repetition of Words: "through strong governmental support in terms of finance, finance and taxation," - The word "finance" is repeated. This can be corrected to "through strong governmental support in finance and taxation."

ü  Misuse of Words: "only in the value chains of high-end industries can survive in the fierce competition," - This phrase is somewhat confusing. It can be clarified by adding a subject: "only businesses in the value chains of high-end industries can survive in the fierce competition."

ü  Conjunction Error: "Innovation ability is the core element of the city to climb to the high parts of value chains, only continuous innovation input and innovation output can control the core parts of the value chains;" - There is a mistake in connecting two independent clauses. You can use a period or an appropriate conjunction, such as: "Innovation ability is the core element of the city to climb to the high parts of value chains. Only continuous innovation input and innovation output can control the core parts of the value chains."

·   Unclear Phrase: "parts of different value chains parts" - This phrase is unclear and can be clarified to "parts of different value chains" or "different segments of value chains." Review statements that may be confusing or unclear and elaborate further if necessary.

·  This section could be enriched with a deeper discussion about what these findings mean for the industry, the regional economy, or this field of research in general.

Conclusion

The conclusion section of this article seems to have successfully outlined the main findings of the research in a structured manner.

Recommendation

This recommendations section has provided a clear view of the steps needed to address the issues identified in the research. However, there is room for improvement in the presentation and connectivity between recommendations and the inclusion of metrics or evaluation methods.

·         Although detailed, these recommendations may be too complex and dense, making it difficult for readers to follow and understand the main points. Grouping the recommendations into broader categories or themes could aid in comprehending and presenting these ideas in a more organized manner.

·   Explaining how these recommendations are interconnected and collectively support the same goal might help readers grasp the bigger picture.

·  There is no guidance on how the effectiveness of these recommendations can be assessed or measured. Providing advice on measuring or evaluating success in implementing these recommendations would add practical value to this section.

· Including some examples or case studies on implementing these recommendations would make this text more accessible and relevant to readers.

Reference

The references are relevant to the research focus on regional labor division in value chains, especially in the context of the electronic information industry and the Yangtze River Delta region. These articles encompass various aspects, including global influences, regional patterns, industrial agglomeration, and specific factors contributing to the dynamics of labor division.

However, there are a few points that may need further consideration:

·   Most of the references focus on China and its surrounding region. If this research aims to be applied or compared globally, it may be necessary to add literature that includes analyses of regional value chains in other areas. 

This research is heavily concentrated on the electronic information manufacturing industry. If the goal is to highlight the specific characteristics of this industry within a regional context, references related to other industries may not be needed. However, if the aim is to take a broader perspective on regional labor division, additional references that include analyses of other industries may be useful.

Moderate editing of the English language is required. Several grammatical and syntactical errors can disrupt the flow of reading. 

Author Response

1Abstract

(1)The authors need to add a little information about this study's background or broader context. An explanation of the importance of this topic in a global or national context might enhance clarity and relevance.

Response:

Thanks for the suggestion. We have added a background context in the first paragraph of the abstract on page 1.

(2)Discussing the practical implications of these findings could provide additional insights into the importance of this research.

Response:

Thanks for the suggestion. We added the practical implications after each research finding.

2Introduction

Response:

Thanks for all the suggestion. We restructured the section by writing the introduction and literature review in two parts. For detailed revised sections, see the pages from 1 to 5.

(1)Although providing much information, this introduction is lengthy and dense. This may overwhelm the reader and can blur the key points. If information is not directly relevant to the research objectives or questions, consider reducing it or moving it to another section of the article.

Response:

Thanks for the suggestion. The authors have rewritten the introduction section, focusing on the basic background, the formulation of the problem, and the importance of this study, so that the reader can have a general perception of the starting point of the research in this paper.

(2)Some sentences may be overly complex or lengthy. They could be divided into shorter sentences without losing meaning.

Response:

Thanks for the comments. We have rewritten the long sentences in the paper for simplicity and clarity.

(3)Some sections might require smoother transitions between points, which would assist in guiding the reader through complex arguments. In this case, some sections might feel abrupt or disconnected. Adding transition sentences between different sections can help guide the reader. For example, after explaining the industrial background, you could add, "In this context, previous research has revealed..." to move the reader to the literature review.

Response:

Thanks for the suggestion. We add transition sentences between different sections.

(4)Ensure that each section follows logically from the one before. If necessary, rearrange some paragraphs or sections to ensure a more logical flow.

Response:

Thanks for the suggestion. We have reorganized and rearranged the logic of the paragraphs in the article.

(5)Check if any sudden changes in style or tone might disturb the reader. Maintain a consistent style and tone throughout all sections.

Response:

Thanks for the suggestion. We read the content of each paragraph in the text and revised the sentences that did not make logical sense accordingly.

Response:

(6)Although the author mentions the importance of this topic in the regional context, adding information about its global and national significance might provide a broader context.

Response:

Thanks for the suggestion. We have added a modification to the first paragraph of the section of introduction on page 2.

3Result and discussion

(1)Some statements may be confusing or unclear, such as the word "finance," which seems to be repeated in the same context.

Response:

Thanks for the comments. We fixed the expression of the word based on specifics.

(2)Several grammatical and syntactical errors can disrupt the flow of reading. Some examples include:

Repetition of Words: "through strong governmental support in terms of finance, finance and taxation," - The word "finance" is repeated. This can be corrected to "through strong governmental support in finance and taxation."

Response:

Thanks for the comments. We deleted first finance, and amend the sentence to "through strong governmental support in finance and taxation."

Misuse of Words: "only in the value chains of high-end industries can survive in the fierce competition," - This phrase is somewhat confusing. It can be clarified by adding a subject: "only businesses in the value chains of high-end industries can survive in the fierce competition."

Response:

Thanks for the suggestion. We modify this sentence“"only in the value chains of high-end industries can survive in the fierce competition" to “only businesses in the value chains of high-end industries can survive in the fierce competition."

Conjunction Error: "Innovation ability is the core element of the city to climb to the high parts of value chains, only continuous innovation input and innovation output can control the core parts of the value chains;" - There is a mistake in connecting two independent clauses. You can use a period or an appropriate conjunction, such as: "Innovation ability is the core element of the city to climb to the high parts of value chains. Only continuous innovation input and innovation output can control the core parts of the value chains."

Response:

Thanks for the suggestion. We modify this sentence "Innovation ability is the core element of the city to climb to the high parts of value chains, only continuous innovation input and innovation output can control the core parts of the value chains;" to "Innovation ability is the core element of the city to climb to the high parts of value chains. Only continuous innovation input and innovation output can control the core parts of the value chains."

Unclear Phrase: "parts of different value chains parts" - This phrase is unclear and can be clarified to "parts of different value chains" or "different segments of value chains." Review statements that may be confusing or unclear and elaborate further if necessary.

Response:

Thanks for the suggestion. We modify this sentence "parts of different value chains parts" to "different segments of value chains." We double-checked the words, grammar and tenses of this paper and made careful modifications on this.

(3)This section could be enriched with a deeper discussion about what these findings mean for the industry, the regional economy, or this field of research in general.

Response:

Thanks for the suggestion. we delete the section of recommendations, and added the section of discussion. The discussion part mainly discusses the advanced nature of VCDI compared with existing research methods, the differences between the results of innovation variables and previous studies, and the importance of strengthening the industrial links between core cities and peripheral cities. The revised section can see on page 19 to 20.

4Conclusion and Recommendation

(1)The conclusion section of this article seems to have successfully outlined the main findings of the research in a structured manner.

Response:

Thanks. We have sorted out and modified the content of the conclusion to make it more smooth.

(2)This recommendations section has provided a clear view of the steps needed to address the issues identified in the research. However, there is room for improvement in the presentation and connectivity between recommendations and the inclusion of metrics or evaluation methods.

Response:

Thanks for the suggestion. we delete the section of recommendations, and added the discussion. In the discussion, we discussed the evaluation method and made some elaboration on the last paragraph in page 20.

(3)Although detailed, these recommendations may be too complex and dense, making it difficult for readers to follow and understand the main points. Grouping the recommendations into broader categories or themes could aid in comprehending and presenting these ideas in a more organized.

Response:

Thanks for the suggestion. we delete the section of recommendations, and added the discussion. At the same time, many changes have been made to the language writing, and strive to be concise and clear, and can clearly express the author's point of view.

(4)Explaining how these recommendations are interconnected and collectively support the same goal might help readers grasp the bigger picture. There is no guidance on how the effectiveness of these recommendations can be assessed or measured. Providing advice on measuring or evaluating success in implementing these recommendations would add practical value to this section. Including some examples or case studies on implementing these recommendations would make this text more accessible and relevant to readers.

Response:

Thanks for mentioning so many good suggestions. After considering the opinions of three experts, as well as the length of the article, we delete the suggestion section and replaced it with the discussion section. Thanks for proposing such good research ideas, we will We will consider it carefully and address it in the future article.

5Reference

(1)Most of the references focus on China and its surrounding region. If this research aims to be applied or compared globally, it may be necessary to add literature that includes analyses of regional value chains in other areas.

Response:

Thanks for the suggestion. We add literatures of regional value chains in other areas.

(2)Additional references that include analyses of other industries.

Response:

Thanks for the suggestion. We add literatures of other industries.

Reviewer 3 Report

A few questions remain about the methodology:

1. Was the localization of the companies according to the registered seat or the city of the plant carrying out the actual activity?

2. What procedure was used to standardize the VCDi variables?

3. Did the value of the standardized variables fall into the interval [0-1] or (0-1)?

4. What software was used for fractional regression modeling?

5. Brief presentation of the Arc GIS spatial classification method

In lines 197-199, something is wrong in terms of language and spelling (beginning of sentences, typos)

Author Response

  1. Was the localization of the companies according to the registered seat or the city of the plant carrying out the actual activity?

Response:

Thanks for this good question. The address data of listed companies are mainly obtained by reviewing the annual reports of the companies to obtain the addresses of their places of operation and mapping them to the cities in which they are located. The data of subsidiaries are also obtained from the data of company annual reports, and are screened according to their specific business activities, and the data of some subsidiaries that are not related to the business activities of electronic information are excluded. So as not to confuse the reader, we have added two footnotes on page 6.

  1. What procedure was used to standardize the VCDU variables?

Response:

Thanks for this question. We use min-max normalization to standardize the VCDI. We added a sentence to explain this problem at first paragraph on page 7.

  1. Did the value of the standardized variables fall into the interval [0-1] or (0-1)?

Response:

Thanks for this question. The VCDI is fall into the interval [0-1], and added a sentence to explain this problem at first paragraph on paper 8.

  1. What software was used for fractional regression modeling?

Response:

Thanks for the suggestion. We use stata14 software to conduct the fractional response regression. In response, we add a sentence in the first paragraph of page 15.

  1. Brief presentation of the Arc GIS spatial classification method

Response:

Thanks for this suggestion. We have added Arc GIS spatial classification method on pages 9 and 10.

Round 2

Reviewer 1 Report

Thank you for considering my recommendations.

There are no comments.

Reviewer 3 Report

Thank you for clarifying the used software, making interval designations equivocal, editing typos, and approaching the requests. I recommend the paper for publication in its current form.